# An Assessment of the Ocular Toxicity of Two Major Sources of Environmental Exposure

**DOI:** 10.3390/ijerph21060780

**Published:** 2024-06-15

**Authors:** Steven H. Rauchman, Lora J. Kasselman, Ankita Srivastava, Joshua De Leon, Allison B. Reiss

**Affiliations:** 1The Fresno Institute of Neuroscience, Fresno, CA 93730, USA; dr.rauchman@yahoo.com; 2Research Institute, Hackensack Meridian Health, Edison, NJ 08502, USA; lora.kasselman@hmhn.org; 3Department of Medicine and Biomedical Research Institute, NYU Grossman Long Island School of Medicine, Mineola, NY 11501, USA; ankita.srivastava@nyulangone.org (A.S.); joshua.deleon@nyulangone.org (J.D.L.)

**Keywords:** toxic chemical, environmental exposure, dry eye disease, Iraq burn pit emissions, ocular surface

## Abstract

The effect of airborne exposure on the eye surface is an area in need of exploration, particularly in light of the increasing number of incidents occurring in both civilian and military settings. In this study, in silico methods based on a platform comprising a portfolio of software applications and a technology ecosystem are used to test potential surface ocular toxicity in data presented from Iraqi burn pits and the East Palestine, Ohio, train derailment. The purpose of this analysis is to gain a better understanding of the long-term impact of such an exposure to the ocular surface and the manifestation of surface irritation, including dry eye disease. In silico methods were used to determine ocular irritation to chemical compounds. A list of such chemicals was introduced from a number of publicly available sources for burn pits and train derailment. The results demonstrated high ocular irritation scores for some chemicals present in these exposure events. Such an analysis is designed to provide guidance related to the needed ophthalmologic care and follow-up in individuals who have been in proximity to burn pits or the train derailment and those who will experience future toxic exposure.

## 1. Introduction

Airborne environmental toxins pose many health risks, and studies of these consequences often focus on the lungs and internal organs and, particularly, on their carcinogenic and mutagenic potential [1,2,3]. However, there is very little literature on the effect of defined, intense exposure to airborne hazards on the eye [4]. The anterior surface of the eye, comprising the cornea and conjunctival tissues, receives direct exposure to the air and airborne chemicals and is vulnerable to injury by these chemicals. 

This paper will focus on two major environmental events in recent history that have had an impact on the ocular health of those in the vicinity. These events are (1) the Ohio train derailment and toxic spill and (2) the burn pits used by the military for waste disposal abroad. The scope and impact of these incidents on the eye is the subject of this work. 

In this study, a screening of the chemicals released into the air from burn pits and from the East Palestine train derailment was conducted using in silico tests in a first attempt to predict the effects on the ocular surface.

There is a significant need to allow computer models to test for chemical ocular toxicity. Such in silico methods provide a low-cost alternative to biological testing that overcomes this price barrier and paves the way for the prediction of the ocular toxicity of chemicals that have not yet been evaluated [5]. Quantitative structure activity (QSAR) can reflect the potential biological activity based on a chemical’s structure [6]. Numerous chemicals are patented each year, and many chemicals are commercially available, but their impact on human organs, including the eyes, is speculative [7]. Traditional Draize tests use exposure of rabbit eyes to a known chemical [8]. This test is expensive, and this method raises serious ethical concerns [9]. In vitro tests are less expensive but still not practical in many instances [10]. Chemicals in the work environment are an established cause of eye irritation and dry eyes. Exposure to organic chemicals in workers in the dry-cleaning industry is known to be associated with symptomatic dry eyes [11]. Dry eye disease with the loss of tear film homeostasis is a painful condition that can cause corneal epithelial damage and visual impairment [12]. It negatively impacts one’s quality of life and imposes a significant economic and social burden globally [13]. Volatile compounds in the atmosphere can directly reach the ocular surface and promote inflammation and cytotoxicity, leading to dry eye disease, pain, and burning [14].

In this paper, new data will be presented on Iraq burn pits and the train derailment in East Palestine, Ohio, USA. This will underscore the underrecognized link between acute and subacute environmental events and long-term damage to the surface of the eye so that actions can be taken to mitigate the risk and consequences for those currently experiencing ocular problems, considering future occurrences. 

## 2. Materials and Methods

Advanced Chemistry Development (ACD) Labs (Toronto, ON, Canada) applied in silico methods to detect the ocular irritation response to air contaminants (https://www.acdlabs.com/products/percepta-platform/tox-suite/, accessed on 16 June 2023). ACD labs employed the chemical structures of 49 chemicals detected at burn pits (full list available upon request) to provide the structure-based calculation of the toxicity endpoints. There was no attempt to analyze the particulate matter or dust as this would not conform to a strict chemical analysis. The general properties were calculated based on experimental data using the ADME Suite and the Tox Suite on the ACD/LAB Percepta platform, Release 2022.2.1 (Build 3577, 7 June 2022) (https://www.acdlabs.com/products/percepta-platform/adme-suite/, accessed on 16 June 2023) [15]. The algorithm for predicting eye irritation in the Tox Suite is based on standard rabbit Draize test data transformed into a binary variable (irritating/not irritating), where the chemicals are considered irritants if they produce at least moderate irritation levels according to the original Draize classification scheme [8,16]. The predictive model was built using structural fragments and physicochemical descriptors, and its output was a probability that the test chemical would cause moderate or severe eye irritation [17]. The experimental data used for the development of the predictive models were sourced from the European chemical substances information system (European Chemical Inventory) and the Registry of Toxic Effects of Chemical Substances (RTECS) databases [18,19]. The final datasets comprised over 2100 molecules for both eye and skin irritation, including qualitative irritation categories determined from tests conducted on adult albino rabbits. Predictive models of the irritation potential were constructed using the binomial Partial Least Squares (PLS) method. These models integrated key physicochemical properties of the chemicals, such as ionization and molecular size, alongside fragmental descriptors, including predefined substructures known to significantly influence the property under analysis. The resulting models demonstrated a high accuracy, with prediction accuracies of 78% for rabbit eye irritation and 73% for rabbit skin irritation. A more thorough analysis of the physiochemical properties determining toxicity is described by Szymański et al. and Nicholas et al. [20,21]. 

## 3. Results

A total of 48 chemicals were analyzed (Table 1) and 3 chemicals—acrolein, chlorodifluoromethane, and methylene chloride—were red-flagged as severe irritants (>0.75 ocular irritation where 1.0 is the maximum). The ACD has since verified that these chemicals were appropriately red-flagged based on their analysis. This does not indicate that the other molecules are non-irritating. However, a high degree of certainty is assigned to red-flagged chemicals.

Figure 1 depicts the chemicals found in Iraqi burn pits predicted to be eye irritants. This provides a quantitative baseline to establish a standard when examining future toxic spills. The ocular toxicity bar was intentionally set high to validate the concept introduced in this study. The burn pit chemicals were chlorodifluormethane (also known as difluoromonochloromethane) with an ocular irritation of 0.77, methylene chloride (also known as dichloromethane) with an ocular irritation of 0.76, and acrolein, with an ocular irritation of 0.8. The specific ocular toxicities are independently referenced [22,23,24,25].

The ACD lab results from the Ohio train derailment in East Palestine were similarly analyzed. Two out of the seven compounds were referenced in the literature [26]. Vinyl chloride was rated 0.76, and separate sources affirm this specific toxicity of vinyl chloride [27,28]. About 1 million pounds of vinyl chloride gas were emitted into the environment following the derailment [29]. Butylacrylate was described to have a very high ocular toxicity, with a rating of 0.86. Figure 2 depicts the chemicals found in the Ohio train derailment predicted to be eye irritants.

Overall, in Ohio, two out of the seven chemicals were identified as probable eye irritants, and, in the Iraqi burn pits, three out of the forty-eight chemicals were identified as probable eye irritants.

## 4. Discussion

The most common manifestation of ocular surface toxic exposure is dry eye disease [30,31]. The public workshop series hosted by the National Academies of Sciences, Engineering, and Medicine in November 2023 named the health research priorities for the East Palestine hazardous materials’ release and named eye irritation as a commonly reported symptom [32]. Sanchez et al. found that veterans exposed to burning oil fields also exhibited more signs of dry eye disease, and smoke from other types of burn and fire situations is also a culprit [33,34] The ophthalmology community has long recognized the importance of dry eye disease as a result of eye irritation, and it is a frequent topic among clinicians and in the ophthalmology literature [35,36,37]. There are multiple FDA-approved prescription drugs and over-the-counter artificial tear formulations designed to treat dry eyes [38,39]. However, the clinical relationship between dry eye disease and specific chemical exposure is not frequently addressed. Interest in exposure-related damage to organs such the lungs, heart, and liver can be severe and easily discernible, while more insidious problems such as ocular surface irritation tend to be of a lower priority. This article is an attempt to focus our attention on this often-overlooked consequence of environmental exposure. 

The context for this discussion is the harm imposed by burn pits and the East Palestine train derailment, so these incidents merit a brief overview. 

Burn pits have been used to dispose of waste predominantly in Iraq [40,41]. Hundreds of thousands of American troops have been exposed to huge amounts of toxic waste incinerated in large open pits with few, if any, environmental safeguards [42]. The VA health system has documented the extent of this exposure [43,44]. The actual list of the chemicals has been documented [24,45]. Ocular surface symptoms have been demonstrated in veterans returning from Iraq [14]. A summary of government data with specific chemicals has been compiled by Makesha Sink OD. These refer to a list of actual chemicals discussed by DOD. Specific ocular irritants are listed [45,46]. This list formed the basis of the data sent to the ACD labs for analysis. While not an exhaustive list, it conforms to what various government agencies have been reviewing.

The East Palestine, Ohio, train derailment took place on the evening of February 3, 2023. It led to a toxic spill and an environmental event [32,47,48]. The industrial chemicals and fire led to a constellation of eye symptoms in both the community and the first responders [49]. These included burning, tearing, and eye pain. An EPA letter listing the specific chemicals to be examined is noted by Jill Newmark in STAT [50]. This list forms the basis of the ACD submission, and, since then, the EPA has compiled its air-sampling data on its website [27,51]. 

The technology to assess ocular surface toxicity is well developed [52,53]. Many previous papers have discussed at length the three main currently available approaches for measuring toxic changes to the surface of the eye [54]. These are (1) the Draize test, (2) in vitro testing, and (3) in silico models [55]. The Draize test requires animal models and is not practical for assessing multiple molecules released in toxic events [8,56]. The in vivo methods are expensive and time consuming [57]. The in silico models are well adapted for a rapid analysis of identified chemicals [58,59]. Of these three, we applied an in silico model for this study as it is practical and widely applied in studies of chemical exposure toxicity [60,61].

Dry eye disease results from inadequate tear production or a low tear quality and causes visual disturbance, eye irritation, and discomfort [62,63]. The diagnosis is made by multiple assessments, with an evaluation of ocular subjective symptoms, tear volume, tear break-up time, tear osmolality, slit lamp examination, and surface staining [64]. Instability of the tear film leads to an inflammatory reaction with oxidative stress and an increase in the concentration of inflammatory cytokines in the tears [65,66]. Inflammation would likely be a key mediator in chemical toxic exposure [55,67].

Ocular surface injury brings about a huge economic burden and a loss of quality of life [68,69]. In 2011, the average cost of managing patients with dry eye was estimated at USD11,302 per patient, with an overall cost of USD55.4 billion to the United States [70]. The expenditure for dry eye medications per patient has been estimated at USD1873 per year in the United States [71]. A retrospective study from China found the burden of dry eye disease on the health care system to be somewhere in the range of USD104.2–USD166.6 billion per year [72]. It contributes to absenteeism, unemployment, and lower productivity [73,74]. Mental health is also negatively influenced by both airborne exposure and dry eye disease [75,76]. Billions of USD in direct and indirect economic costs have been documented [77,78]. 

Despite its enduring health consequences, dry eye disease is generally not factored into the long-term analysis of important chemical toxic exposure, and the clinical relationship between dry eye disease and specific chemical exposure is rarely addressed. In fact, a recent review of burn pit-associated symptoms in veterans did not consider the eyes [79]. A clinical ophthalmologist frequently encounters and manages dry eye disease when a patient presents in the office setting. The evaluation, categorization, and treatment options for dry eye disease are well characterized in the literature, but the condition can have a highly variable constellation of symptoms and is, therefore, often undiagnosed, and treatment opportunities are missed [80,81]. The first option usually involves the regular instillation of over-the-counter lubricants in the form of artificial tears [82]. The control of the inflammation with topical corticosteroids, cyclosporine A, or oral derivatives of tetracycline may offer relief. It is important that acute chemical events should be included among the differential causes of dry eye disease. If the onset of the symptoms coincides with such an acute event or environmental exposure, then this issue is relevant not only to the patient, but to those managing cleanup, countermeasures, and data collection for the mitigation of health consequences [83].

Major environmental disasters have occurred and will likely continue to occur with a reasonable frequency. As an example, the World Trade Center disaster on 11 September 2001 led to many health consequences as a result of toxic debris, including eye irritation, but this was understandably overshadowed by a myriad of other health problems [84,85]. The most pressing concern is to remove individuals from immediate harm [86]. There is an emergency response system in place to address such concerns [87]. Secondary response is important but may be muted as other issues distract attention [88,89]. Proactive recognition may avert a repeat of similar disasters, and the recognition of the risk to the ocular surface can prompt not only timely treatment, but adequate surveillance and the development of improved protocols to minimize eye damage [90,91]. Unfortunately, in many parts of the world, open burning as a means of waste disposal is common, and exposure is ongoing and unavoidable without awareness and substantial change [92,93]. 

This study presents some limitations. There can be great variability in the chemicals released from different types of burn sites, so the chemicals detected here may not be representative of burn sites in general. This was an in silico test, so even though some of the chemicals have been shown by independent means to negatively impact human health, substantiating the evidence through future data collection would support our contention that these chemicals are toxic to the eyes in real-life scenarios [94]. 

## 5. Conclusions

This study presents new ocular toxicology data for two major events: Iraqi burn pits and the Ohio train derailment. Our data demonstrate the application of easily accessible and proven scientific measurements to determine the trajectory and significance of toxic exposure that will undoubtedly occur in the future. This work can provide a path for the institution of safety measures and monitoring protocols where individuals exposed to surface ocular toxins can be carefully followed for the development of chronic ocular surface disease or dry eye disease. Appropriate sampling and analysis are the necessary first steps in defining the toxins released. Proactive mitigation and education programs by community-based, academic, and governmental organizations can improve the outcomes. 

The eyes are directly exposed to any toxic chemicals released into the air. The immediate consequences are usually obvious, with significant burning and tearing. The long-term impact of exposure is an important knowledge gap, but it is known that dry eye disease brings enormous personal and economic consequences. Our in silico analysis provides quantitative data, and it is our hope that it will be applied as a tool to study the next toxic spill.

## Figures and Tables

**Figure 1 ijerph-21-00780-f001:**
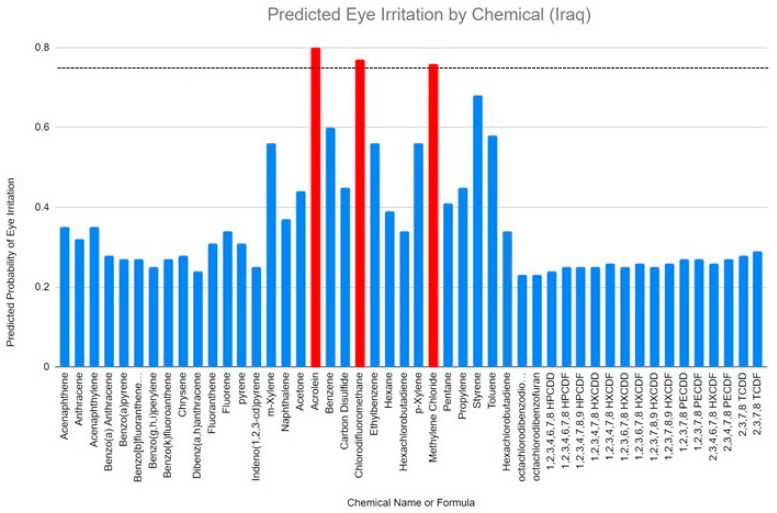
Bar graph of predicted eye irritation by chemicals in Iraqi burn pit. The chemicals with a predicted eye irritation of 0.75 or greater are indicated in red. The dashed line represents a predicted toxicity threshold of 0.75.

**Figure 2 ijerph-21-00780-f002:**
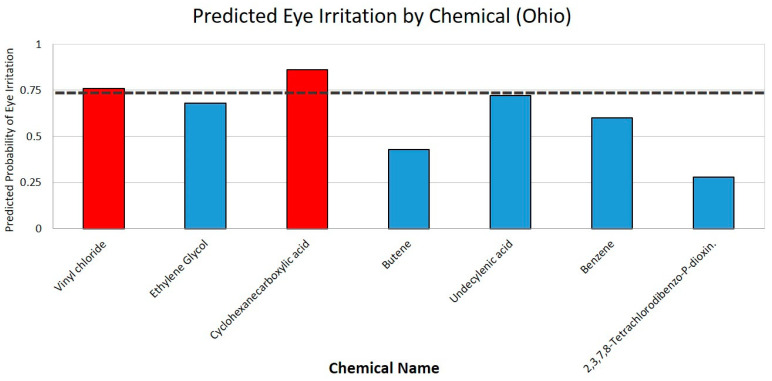
Bar graph of predicted eye irritation by chemicals in Ohio. The chemicals with a predicted eye irritation of 0.75 or greater are indicated in red. The dashed line represents a predicted toxicity threshold of 0.75.

**Table 1 ijerph-21-00780-t001:** Chemicals detected at burn pits with their predicted probability of ocular irritation.

ID	Name	SMILES	Probability of Eye Irritation	Chemical Formula
1	Acenaphthene	C1CC2=CC=CC3=C2C1=CC=C3	0.35	C12H10
2	Anthracene	C1=CC=C2C=C3C=CC=CC3=CC2=C1	0.32	C14H10
3	Acenaphthylene	C1=CC2=C3C(=C1)C=CC3=CC=C2	0.35	C12H8
4	Benzo(a) Anthracene	C1=CC=C2C(=C1)C=CC3=CC4=CC=CC=C4C=C32	0.28	C18H12
5	Benzo(a)pyrene	C1=CC=C2C3=C4C(=CC2=C1)C=CC5=C4C(=CC=C5)C=C3	0.27	C20H12
6	Benzo[b]fluoranthene-d12	C1=CC=C2C3=C4C(=CC=C3)C5=CC=CC=C5C4=CC2=C1	0.27	C20D12
7	Benzo(g,h,i)perylene	C1=CC2=C3C(=C1)C4=CC=CC5=C4C6=C(C=C5)C=CC(=C36)C=C2	0.25	C22H12
8	Benzo(k)fluoranthene	C1=CC=C2C=C3C4=CC=CC5=C4C(=CC=C5)C3=CC2=C1	0.27	C20H12
9	Chrysene	C1=CC=C2C(=C1)C=CC3=C2C=CC4=CC=CC=C43	0.28	C18H12
10	Dibenz(a,h)anthracene	C1=CC=C2C(=C1)C=CC3=CC4=C(C=CC5=CC=CC=C54)C=C32	0.24	C22H14
11	Fluoranthene	C1=CC=C2C(=C1)C3=CC=CC4=C3C2=CC=C4	0.31	C16H10
12	Fluorene	C1C2=CC=CC=C2C3=CC=CC=C31	0.34	C13H10
13	pyrene	C1=CC2=C3C(=C1)C=CC4=CC=CC(=C43)C=C2	0.31	C16H10
14	Indeno(1,2,3-cd)pyrene	C1=CC=C2C(=C1)C3=C4C2=CC5=CC=CC6=C5C4=C(C=C6)C=C3	0.25	C22H12
15	m-Xylene	CC1=CC(=CC=C1)C	0.56	C8H10
16	Naphthalene	C1=CC=C2C=CC=CC2=C1	0.37	C10H8
17	Acetone	CC(=O)C	0.44	C3H6O
18	Acrolein	C=CC=O	0.8	C3H4O
19	Benzene	C1=CC=CC=C1	0.6	C6H6
20	Carbon Disulfide	C(=S)=S	0.45	CS_2_
21	Chlorodifluoromethane	C(F)(F)Cl	0.77	CHClF2
22	Ethylbenzene	CCC1=CC=CC=C1	0.56	C8H10
23	Hexane	CCCCCC	0.39	C6H14
24	Hexachlorobutadiene	C(=C(Cl)Cl)(C(=C(Cl)Cl)Cl)Cl	0.34	C4Cl6
25	p-Xylene	CC1=CC=C(C=C1)C	0.56	C8H10
26	Methylene Chloride	C(Cl)Cl	0.76	CH2Cl2
27	Pentane	CCCCC	0.41	C5H12
28	Propylene	CC=C	0.45	C3H6
29	Styrene	C=CC1=CC=CC=C1	0.68	C8H8
30	Toluene	CC1=CC=CC=C1	0.58	C6H5CH3
31	Hexachlorobutadiene	C(=C(Cl)Cl)(C(=C(Cl)Cl)Cl)Cl	0.34	C4Cl6
32	Octachlorodibenzodioxin	C12=C(C(=C(C(=C1Cl)Cl)Cl)Cl)OC3=C(O2)C(=C(C(=C3Cl)Cl)Cl)Cl	0.23	C12H4Cl4O2
33	Octachlorodibenzofuran	C12=C(C(=C(C(=C1Cl)Cl)Cl)Cl)OC3=C2C(=C(C(=C3Cl)Cl)Cl)Cl	0.23	C12Cl8O
34	1,2,3,4,6,7,8 HPCDD	C1=C2C(=C(C(=C1Cl)Cl)Cl)OC3=C(O2)C(=C(C(=C3Cl)Cl)Cl)Cl	0.24	C12HCl7O2
35	1,2,3,4,6,7,8 HPCDF	C1=C2C3=C(C(=C(C(=C3Cl)Cl)Cl)Cl)OC2=C(C(=C1Cl)Cl)Cl	0.25	C12HCl7O
36	1,2,3,4,7,8,9 HPCDF	C1=C2C(=C(C(=C1Cl)Cl)Cl)C3=C(O2)C(=C(C(=C3Cl)Cl)Cl)Cl	0.25	C12HCl7O
37	1,2,3,4,7,8 HXCDD	C1=C2C(=CC(=C1Cl)Cl)OC3=C(O2)C(=C(C(=C3Cl)Cl)Cl)Cl	0.25	C12H2Cl6O2
38	1,2,3,4,7,8 HXCDF	C1=C2C(=CC(=C1Cl)Cl)OC3=C2C(=C(C(=C3Cl)Cl)Cl)Cl	0.26	C12H2Cl6O
39	1,2,3,6,7,8 HXCDD	C1=C2C(=C(C(=C1Cl)Cl)Cl)OC3=CC(=C(C(=C3O2)Cl)Cl)Cl	0.25	C12H2Cl6O2
40	1,2,3,6,7,8 HXCDF	C1=C2C3=C(C(=C(C=C3OC2=C(C(=C1Cl)Cl)Cl)Cl)Cl)Cl	0.26	C12H2Cl6O
41	1,2,3,7,8,9 HXCDD	C1=C2C(=C(C(=C1Cl)Cl)Cl)OC3=C(C(=C(C=C3O2)Cl)Cl)Cl	0.25	C12H2Cl6O2
42	1,2,3,7,8,9 HXCDF	C1=C2C(=C(C(=C1Cl)Cl)Cl)C3=C(C(=C(C=C3O2)Cl)Cl)Cl	0.26	C12H2Cl6O
43	1,2,3,7,8 PECDD	C1=C2C(=CC(=C1Cl)Cl)OC3=C(C(=C(C=C3O2)Cl)Cl)Cl	0.27	C12H3Cl5O2
44	1,2,3,7,8 PECDF	C1=C2C(=CC(=C1Cl)Cl)OC3=CC(=C(C(=C23)Cl)Cl)Cl	0.27	C12H3Cl5O
45	2,3,4,6,7,8 HXCDF	C1=C2C3=CC(=C(C(=C3OC2=C(C(=C1Cl)Cl)Cl)Cl)Cl)Cl	0.26	C12H2Cl6O
46	2,3,4,7,8 PECDF	C1=C2C3=CC(=C(C(=C3OC2=CC(=C1Cl)Cl)Cl)Cl)Cl	0.27	C12H3Cl5O
47	2,3,7,8 TCDD	C1=C2C(=CC(=C1Cl)Cl)OC3=CC(=C(C=C3O2)Cl)Cl	0.28	C12H4Cl4O2
48	2,3,7,8 TCDF	C1=C2C3=CC(=C(C=C3OC2=CC(=C1Cl)Cl)Cl)Cl	0.29	C12H4Cl4O

Abbreviations. Simplified molecular input line entry system (SMILES).

## Data Availability

The datasets analyzed as part of the current study can be made available by the corresponding author upon reasonable request.

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
