# Peer review of "An Assessment of the Ocular Toxicity of Two Major Sources of Environmental Exposure"

_ijerph, 2024, doi:10.3390/ijerph21060780_

Round 1

Reviewer 1 Report (Previous Reviewer 1)

Comments and Suggestions for Authors

The authors have managed to address my comments and submitted as Communication.

Reviewer 2 Report (Previous Reviewer 2)

Comments and Suggestions for Authors

It can be accepted in its current form as a communication paper.

Reviewer 3 Report (Previous Reviewer 4)

Comments and Suggestions for Authors

There are no comments

This manuscript is a resubmission of an earlier submission. The following is a list of the peer review reports and author responses from that submission.

Round 1

Reviewer 1 Report

Comments and Suggestions for Authors

In this work, Rauchman and co-workers utilized in silico methods to assess ocular toxicity from airborne exposures in Iraqi burn pits and the East Palestine train derailment. Their findings underscore the importance of ophthalmologic care for individuals exposed to such events, guiding future healthcare interventions. This study highlights the potential long-term impact on ocular health and emphasizes the need for proactive measures in managing these exposures.

It is an interesting topic. However, it is a poorly written manuscript. Below are my major comments.

1) The introduction section is very poor. Please provide a page of background information. Focus more on pathogenesis of ocular diseases and how these airborne exposures are potential risk factors.

2) I didn't find anything meaningful in Materials and Methods section. The authors should refer other research papers to get an idea. 

3) The results are not informative. 

4) Overall, this cannot be published as a research article. Based on its very very limited data and less information, this can be published as a communication. Even before that, the above mentioned points must be significantly addressed. 

Author Response

Replies in file below

Reviewer 2 Report

Comments and Suggestions for Authors

Using model simulations, Rauchman et al. studied the potential irritating influences of various chemicals on the ocular surface. I suggest rejection of this work in its current work. Below are my reasons:

This manuscript is more like a lab report. It contains no in-depth discussions of the results. The introduction part did not summarize and discuss previous findings. Instead, most references were cited in the Discussion part, where they should discuss their own results. They showed their results without discussing them.

Some specific comments:

1.      L66: What does ‘ADP’ stand for?

2.      Table 1: ‘SMILES(1)’ is no different from ‘SMILES’. It should be deleted to avoid redundancy. The author should add the chemical formula to Table 1.

3.      Because Table 1 contains all the information provided by Table 2, Table 2 should be deleted. The authors can simply add the names of these three chemicals in the text.

4.      In Figure 1, ‘Acrolein’ appeared twice. Figure 1 just plots values listed in Table 1. It would be more reader-friendly to keep Figure 1 and move Table 1 to the Supplementary Information.

5.      L84: It doesn’t make sense to calculate the percent value when you only have 7 chemicals for Ohio. Also, as indicated in L64, it should be 3 chemicals that were irritants.

6.      Figure 2: I don’t understand why the authors use chemical names in Figure 1 and chemical formula in Figure 2. Is it because the chemicals in Figure 2 are in Table 1 or Figure 1 too?

Author Response

Replies in file below

Reviewer 3 Report

Comments and Suggestions for Authors

Rauchman and colleagues propose a potentially intersting paper about air contaminats and their ocular effect. Nevertheless, there are some major issues:

1) the introduction is too short. The possible general effects on other systems of air contaminants should be summarised. Furthermore, a few lines on dry eye etiology and management must be added in the introduction;

2) the methods section must be enlarged explaining more clearly how the authors determined the ocular toxicity; 

3) the economic burden on pubblic health should be discussed;

4) the practical implications for physicians may also be discussed. 

Author Response

Replies in file below

Reviewer 4 Report

Comments and Suggestions for Authors

The manuscript entitled “An Assessment of the Ocular Toxicity of Two Major Environmental Exposures” studied the impact of environmental exposures from burn pits or train derailments on the ocular surface using in-silico analysis.  It is a good concept and a well-written manuscript with updated references. The summary translates the text well. The introduction has elements that integrate the work, allowing us to evaluate the context in which the manuscript is inserted. The methodology is suitable for what it proposes to evaluate. The results are clear. The discussion is quite complete and alludes to literature references. The conclusions are consistent with the evidence and address the main concept of the review article. In general, the review article represents a quality with good potential for applicability. However, minor comments are suggested below:

Comments:

1.     More explanation in the Introduction section should be added. The introduction is too short.

2.     In Fig 2, Line 92, “Dashed line represents predicted toxicity threshold of 0.75.” It is not a dashed line, please correct it.

3.     In Fig 2, in chemical formula, the numbers should be written in a subscript form for example C11H20O2

4.     Some abbreviations are missed

5.     The references are appropriate.

Author Response

Replies in file below
